# Investigating Older Adults’ Response to Climate Change

**DOI:** 10.3390/ijerph22020154

**Published:** 2025-01-24

**Authors:** Qing Ni, Hua Dong, Antonios Kaniadakis, Zhili Wang, Chang Ge

**Affiliations:** 1Brunel Design School, Brunel University of London, Uxbridge UB8 3PH, UK; 2Department of Computer Science, College of Engineering, Design and Physical Sciences, Brunel University of London, Uxbridge UB8 3PH, UK; antonios.kaniadakis@brunel.ac.uk; 3College of Arts and Media, Tongji University, Shanghai 200070, China; wangzhili@tongji.edu.cn; 4School of Art and Design, Xi’an University of Technology, Xi’an 710048, China; gechang@xaut.edu.cn

**Keywords:** climate change, older adults, low-carbon

## Abstract

Older adults are both vulnerable to the impacts of climate change and uniquely positioned to contribute to climate action. However, their ability and willingness to engage vary significantly due to health disparities, financial constraints, and cultural factors. To ensure inclusivity, climate policies must reflect these differences and empower older adults to participate effectively. This research focused on 30 London-based older adults aged 60–85 (18 women, 12 men) selected via purposive sampling and stratified by gender and climate awareness. All participants were interested in climate issues and engaged in seven small focus groups facilitated by two researchers. Discussions addressed climate perceptions, low-carbon behaviors, and policy recommendations. The findings reveal that tailored communication, featuring simplified language and visual materials, resonates deeply with older adults, fostering better understanding and emotional connection. Participants highlighted practical low-carbon actions they already undertake, such as energy conservation, food waste reduction, and public transport use. They also proposed innovative strategies for promoting climate awareness, including integrating cultural and emotional elements, encouraging intergenerational learning, and providing economic incentives for green practices. To enhance the engagement of older adults in climate action, governments and corporations should develop inclusive communication strategies, provide financial support for adopting green technologies, and foster intergenerational collaboration to share knowledge and experiences in the communities. This study amplifies the voices of older adults in climate discourse, offering actionable insights for shaping communication strategies and policies. While this study provides valuable insights into older adults’ contributions to climate action, future research could expand the sample size and geographic diversity to enhance the generalizability of findings.

## 1. Introduction

The older population is particularly vulnerable to climate change risks, such as extreme weather and resource shortages. Nonetheless, they hold the capacity to play an active role in addressing these challenges. By designing inclusive climate policies, it is possible to mitigate their vulnerabilities while unlocking their potential for contributing to community resilience and broader climate action [1]. While much of the research addressing climate change focuses on individual responsibility, population samples are often generalized, with relatively few studies centered on the perspectives of older people in relation to climate change and personal behavior [2,3,4]. Additionally, the global climate movement, dominated by young people, has spoken of the “failure of older generations to take sufficient action”. This has influenced perceptions of the role of older generations in climate change [5]. Carbon emissions associated with older populations can be influenced by several aspects of lifestyle and socioeconomic conditions, such as energy use within households, transportation choices, and everyday consumption behaviors. Research in Europe indicates that certain lifestyle patterns among older adults, such as residing in larger homes, may result in higher household energy consumption, contributing to carbon emissions [6]. However, other studies suggest that lifestyle adjustments in later life, including reduced mobility or energy use, may lead to lower carbon outputs [7]. For example, health concerns can limit mobility, leading to reduced travel-related emissions in certain areas.

The UK government’s net-zero strategy, aimed at achieving carbon neutrality by 2050 through comprehensive decarbonization policies, introduces both challenges and opportunities for older adults [8]. They can play an active role in this transition by cutting energy consumption and driving community-led adaptation efforts. A 2024 study found that younger people are more likely than older adults to believe in the attainability of the net-zero target [9]. Moreover, the public perceives that white and younger individuals are more likely to benefit from net-zero policies compared with ethnic minorities and older populations [10]. The factors influencing climate change extend across cultural, personal, religious, gender, age, racial, social, corporate, and governmental dimensions [11]. Urbanization, for instance, is a significant driver of carbon emissions, while population aging in urban areas presents unique challenges in addressing climate vulnerability. The impacts of climate change on society’s most vulnerable groups, including older adults, continue to be a focus of academic consideration [12]. While some scholars have sought to mobilize older adults in response to climate change [13], most literature remains centered on the vulnerability and health issues faced by this demographic, without fully recognizing their potential creativity in addressing climate change [14]. Older adults often have more time to engage in civic activities and volunteer services and may possess valuable life experiences and expertise [15]. Additionally, they frequently hold positions of esteem and influence within their communities, as evidenced by their participation in local councils, volunteer services, and leadership in faith-based organizations [16]. Encouraging broader participation from older adults in climate change initiatives could yield tangible benefits at the individual, regional, and national levels [15]. Therefore, climate policy and adaptation strategies must address the needs of older populations to help them better cope with the impacts of climate change while recognizing their pivotal role in society. As a result, case studies focusing on older adults as active participants in developing local policies or projects aimed at climate adaptation are becoming more prevalent in community planning and public policy [13].

This study, conducted in the UK, focuses on older adults residing in London. Through focus group interviews, it delves deeply into their perceptions, behaviors, and recommendations regarding climate action. The qualitative data collected from these discussions will be analyzed using thematic analysis, a robust method for identifying and interpreting patterns within qualitative data. This approach ensures that the insights gained reflect the diverse experiences and perspectives of participants. As long-term residents of the UK, they also shared their expectations regarding the roles of the UK government, society, and businesses in addressing climate change through awareness and tangible actions. Based on the findings of this study, we propose future communication strategies for climate action. These strategies include simplifying technical climate concepts using accessible language and visual materials, fostering intergenerational dialogues to enhance mutual learning and collaboration, integrating cultural and emotional elements to increase engagement, and providing economic incentives for sustainable behaviors. Stakeholder opportunities such as creating community-based programs targeted education campaigns, and more inclusive policy consultations are emphasized as ways to enhance the engagement of older adults in climate resilience, mitigation, adaptation, and education.

Encouraging broader participation from older adults in climate change initiatives could yield tangible benefits at the individual, regional, and national levels. For example, older adults’ involvement in local sustainability projects can inform community-specific adaptation strategies, while their advocacy can shape policy priorities at the national level.

## 2. Research Background

### 2.1. Climate Change and Low-Carbon Behavior

Climate change refers to the long-term alteration of Earth’s temperature, weather patterns, and atmospheric conditions. Climate change is influenced by both natural and anthropogenic factors. Natural drivers, such as volcanic eruptions, solar activity, and orbital variations, have shaped Earth’s climate over millennia. However, the rapid warming observed since the Industrial Revolution is primarily attributed to anthropogenic greenhouse gas emissions, with natural factors playing a smaller, modulating role. Understanding this interaction is crucial for designing effective mitigation strategies. Addressing climate change requires reducing greenhouse gas emissions, transitioning to renewable energy, and adopting sustainable practices to both mitigate and adapt to its impacts. Changes in individual behavior are crucial for mitigating climate change. While the majority of carbon emissions are driven by fossil fuel consumption and large-scale corporate activities, individual behavioral changes can still play a critical role in addressing the climate crisis [17]. Behavioral and social changes to achieve ‘zero emission’ can be guided by two interdependent strategies [18]: (1) enabling individuals to take meaningful action to significantly reduce emissions, and (2) creating an environment that supports public participation in climate action.

Low-carbon behavior can be categorized into three main areas: transport, housing, and food [19]. 1. **Transport:** Emissions can be reduced by limiting travel, particularly long-haul flights; promoting active travel such as cycling and walking; and increasing the use of public transport. Further measures include improving vehicle efficiency and adopting electric vehicles (BEVs). 2. **Housing:** Low-carbon heating, renewable energy adoption, and energy-efficient construction can significantly lower emissions. Reducing living space is primarily related to housing choices while adjusting room temperatures is better categorized under energy efficiency practices within housing management. 3. **Food:** Adopting a vegetarian or vegan diet can lead to significant carbon reductions, along with consuming locally grown food, improving cooking efficiency, reducing waste, and avoiding overconsumption. Governments can also play a key role by implementing measures that support low-carbon behavior. These include 1. institutional empowerment: providing people with the necessary tools and information to drive change, and 2. affordability: ensuring low-carbon products and services are financially accessible [20].

Research highlights that older adults often exhibit a strong environmental concern, commonly characterized as prosocial behavior, in the context of addressing climate change [21]. This prosocial outlook is rooted in their altruistic tendencies and is often guided by a focus on emotionally meaningful relationships and deeper connections with family and friends [22]. Their concern for future generations, driven by a sense of legacy and generativity, further motivates their engagement in actions that benefit others and the environment. These prosocial tendencies are reflected in behaviors such as fostering social bonds, caring for others, and offering support within close-knit social networks [22]. This generational concern can extend to climate action, where older adults may adopt low-carbon behaviors like using energy-saving technologies or participating in sustainable travel options to ensure a better future for their descendants [23]. With adequate support from policies and society, this intrinsic motivation can be harnessed to further engage older adults in meaningful and impactful climate actions.

### 2.2. Aging and Climate Change

#### 2.2.1. Climate Risks to Older People

From a global perspective, climate change presents significant risks to older adults, and these risks are likely to intensify in the context of aging societies. The older population may face several challenges related to climate change, including 1. increased vulnerability to extreme weather events, leading to illness or limited mobility [24]; 2. economic difficulties, as older adults may experience reduced incomes after retirement, leaving them more vulnerable to resource shortages caused by climate-related disasters [25]; and 3. social isolation, which could be exacerbated by extreme weather and natural disasters, as older adults are often more prone to isolation [1].

In the UK, heatwaves pose significant health risks to older adults, with extreme temperature events contributing to excess mortality among this vulnerable population [26]. Data from the Office for National Statistics (ONS, 2021) further support this, showing that older adults face disproportionately higher mortality rates during extreme climate events [27]. Beyond these immediate risks, climate-driven migration places additional stress on family and social support networks [28], while the intersection of an aging population and climate change is expected to increase public health expenditure and exacerbate health and social inequalities [29].

Recognizing older adults solely as a vulnerable group overlooks their potential as active contributors to climate action. As the global older population continues to grow, it is essential to integrate their insights and actions into climate strategies. UNDESA’s aging reports highlight the importance of viewing older adults as key actors in addressing climate challenges [30]. Pillemer et al. (2022) further emphasize the benefits and challenges associated with older adults’ participation in environmental activities [31]. Similarly, Age UK’s “Parliamentary Briefing: The Energy Crisis and Supporting Older People This Winter” (2021) stresses the need for targeted support to help older adults navigate energy crises and transition to sustainable energy practices [32]. The “Healthy Ageing in a Changing Climate” report underscores the importance of involving older adults in climate action through community education and improved resource access, advocating for inclusive, age-friendly strategies to empower older populations in addressing climate challenges [33].

While initiatives like the UK government’s Green Homes Grant have aimed to improve energy efficiency in older households, significant gaps remain [34]. Older adults lack a strong voice in climate discourse, and there is a shortage of communication strategies tailored to their needs [28]. Furthermore, their low participation in environmental actions and the lack of attention to their specific requirements in climate adaptation policies and technological innovations exacerbate existing social inequalities [13].

#### 2.2.2. Carbon Emissions from Older Adults

Older adults have largely been overlooked in climate change discussions, creating a critical knowledge gap regarding their role in addressing the climate crisis [13]. The carbon footprint of older people is closely linked to their lifestyle [35], consumption patterns [36], and health needs [37], with several studies offering different perspectives on this issue.

On the one hand, older adults’ carbon emissions are influenced by both individual and collective characteristics: 1. **Healthcare and energy use:** Older people’s consumption patterns, including higher reliance on healthcare services and medical equipment, are influenced by societal systems such as healthcare accessibility and urban planning. For instance, dispersed urban layouts with limited public transport options often necessitate greater personal vehicle use among older adults, contributing to higher transport-related emissions. Additionally, energy-intensive healthcare services may exacerbate their overall carbon footprint. In addition, many live in larger homes, where heating and maintenance can result in higher energy consumption. Spending more time at home also leads to greater household energy use, such as for heating, cooling, and lighting [7]. 2. **Diet:** Some older adults prefer traditional diets rich in meat, which typically have a higher carbon footprint than plant-based diets. However, increasing awareness of sustainable eating has led some older adults to adopt vegetarian or locally sourced food options, thereby reducing their carbon footprint [38]. 3. **Income and consumption:** The aging population’s consumption patterns are also shaped by changes in income. Following retirement, older people often experience a sharp decline in income, prompting more cautious consumption behavior [39]. 4. **Lifestyles:** Older adults’ established habits and routines can pose challenges to adopting new, long-term strategies for reducing carbon emissions. For example, older adults in Europe tend to rely more on personal vehicles for daily errands and social visits, particularly in rural or suburban areas where public transport options are limited [40]. In contrast, older adults in Asia are more likely to use public transport or walk for similar activities, especially in urban environments with well-developed infrastructure [41]. These differences highlight how individual behaviors are shaped by cultural and regional contexts, underscoring the need for further exploration of these patterns to design tailored, low-carbon interventions [17]. 5. **Climate advocacy and intergenerational responsibility:** Many older adults are actively engaging in climate advocacy, driven by a sense of intergenerational responsibility to ensure a healthier planet for future generations. Organizations like Elders Climate Action and Raging Grannies International highlight the unique role older adults play in promoting sustainability and fostering intergenerational dialogue [42,43].

On the other hand, people’s worldviews and values can be resistant to change [44]. Different generations, shaped by their unique social experiences, may have varied responses to carbon emissions [45]. For instance, those who lived through the Second World War may exhibit frugal habits due to the resource scarcity experienced during that time. However, this does not apply uniformly, as cultural, economic, and geographic factors significantly shape individual behaviors within the same cohort. Similarly, the post-war "baby boomer" generation is diverse, with varying degrees of consumption influenced by regional and socioeconomic contexts. Moreover, as living standards have risen and personal wealth has accumulated, an increasing number of older adults are investing in their health or leisure travel [46].

To reconcile the seemingly contradictory findings of high and low carbon emissions among older adults, a life course perspective can be employed [47]. While older adults may have reduced transport-related emissions due to decreased mobility, their emissions from housing and healthcare tend to be higher. This duality highlights the need for targeted interventions that address specific sources of emissions without overlooking their broader lifestyle patterns.

#### 2.2.3. Engaging Older People in Climate Action Dialogue

Older people have both opportunities and challenges in engaging in climate action. First of all, research has shown that many older individuals are often willing to make significant sacrifices to ensure a healthy climate for future generations [48]. Second, having lived through varied social and environmental changes, older people share a collective understanding of how to adapt to shifting circumstances. Their collective experiences can offer deeper insights into the complexity of climate issues, and older people often approach problem-solving with practicality and a focus on feasible solutions. This makes them more likely to reach a consensus and push for actionable climate policies [17]. In addition, older adults often have more discretionary time compared with younger generations, providing them with unique opportunities to actively participate in climate action. Their availability to engage in activism and advocacy represents an underutilized resource in the fight against climate change [13]. Older adults can contribute meaningfully as community leaders, mentors, and activists in climate action initiatives. Their roles may include organizing local environmental events, advocating for policy changes, and sharing traditional ecological knowledge with younger generations to foster intergenerational collaboration [49]. Additionally, as preservers of traditional ecological knowledge, they serve as vital conduits for passing down sustainable practices, such as composting and planting techniques, within their communities [13].

However, there are notable barriers to older adults’ involvement in climate change actions: 1. **Lack of inclusion:** Despite their potential to contribute meaningfully to climate action, older adults are often insufficiently included in climate-related discussions and decision-making processes. While they are more likely to vote and many older leaders are involved in climate policymaking, these do not necessarily reflect the diverse voices and needs of the older population as a whole. This lack of inclusion is further perpetuated by societal structures, such as ageism and hierarchical power dynamics, which often marginalize older voices in policymaking and public discussions [50]. 2. **Knowledge gaps:** Many older people feel they lack the specialized knowledge needed to contribute effectively to climate action. There is currently a shortage of accessible climate change education specifically tailored for older adults. Tailored education for older adults can include interactive workshops, visually engaging materials, and accessible digital tools designed to cater to their unique learning needs. Programs designed in collaboration with local communities can ensure cultural relevance and address specific barriers, such as limited digital literacy or mobility constraints. This lack of knowledge, combined with a lack of confidence in their ability to engage with complex issues, can prevent older adults from taking action [51]. 3. **Limited opportunities for participation:** Access to participation in climate action is often restricted for older adults. Many initiatives rely on online recruitment and participation, which can be difficult for older individuals who may not be as proficient in digital technology. Targeted digital literacy programs can help bridge this gap by teaching essential digital skills, such as accessing online resources, using mobile apps, and participating in virtual events. Furthermore, many climate action activities tend to cater to younger or professional demographics, overlooking the unique needs and capabilities of older people. By combining digital resources with offline engagement opportunities, such as in-person workshops or local community events, participation can be made more accessible and inclusive for older populations [51]. 4. **Lack of adaptation to older people’s life stage:** Many climate initiatives are not designed with the specific needs of older people in mind. For example, older adults may face mobility issues or reading difficulties, and there are limited climate action activities that accommodate these challenges by reducing physical demands or employing simple technological tools [13]. 5. **Systemic barriers to older adults’ climate action:** Older adults face significant systemic barriers to participating in climate action, including the lack of age-friendly design in urban planning [52], government policies that fail to address their specific needs [53], the high costs of low-carbon technologies [54], and insufficient community support [55]. These barriers reduce their ability to engage in sustainable practices and community initiatives. 6. **Intersecting barriers in climate participation:** Barriers to participation, such as knowledge gaps and limited opportunities, intersect with gender [56], socioeconomic status [57], and cultural backgrounds [58]. For instance, older women from lower-income households may face compounded challenges due to restricted access to resources and societal expectations regarding their roles. Similarly, cultural differences can shape how older adults perceive and engage with climate action.

Involving older adults in climate mobilization can deliver significant benefits across personal, community, and global dimensions. At the individual level, participation in climate action alleviates loneliness, restores a sense of purpose, and boosts self-esteem. Daniel Katey and Senyo Zanu (2024) argue for empowering older adults as climate activists within their communities, enabling them to contribute their expertise—such as knowledge in agriculture or ecological conservation—to climate adaptation efforts [59]. This not only grants them societal recognition but also promotes physical well-being through active engagement [60].

At the community level, older adults’ wealth of knowledge and practical wisdom can catalyze change by supporting both individual and collective environmental efforts [61]. They act as key agents in sharing knowledge and fostering intergenerational collaboration, thereby strengthening community cohesion and ensuring the transmission of valuable insights [62]. Furthermore, their participation in climate mobilization draws attention to the specific needs of vulnerable groups, advancing fairness in policy development [63].

On a global scale, older adults’ historical understanding and long-term perspectives offer unique insights into climate action. Their influence as voters and community leaders highlights the importance of integrating their perspectives into climate adaptation strategies. Moreover, their active participation in global climate negotiations represents the voices of vulnerable populations and underscores the critical need for social climate justice [64].

#### 2.2.4. The Main Theories Related to Low-Carbon Behavior and Environmental Awareness of Older People

Older adults’ low-carbon behaviors and environmental awareness can be explained through several theoretical frameworks. **Rational choice theory** [65] highlights how cost–benefit analysis influences decisions, while **social cognitive theory** [66] focuses on social learning and self-efficacy, with older adults often inspired by observing others. **Value-belief-norm theory** [67] emphasizes the role of moral responsibility, particularly towards future generations, in shaping sustainable actions. The **biopsychosocial model** [68] underscores the impact of health, attitudes, and social support, where mobility limitations or family encouragement affect behaviors. Lastly, **social-ecological systems theory** [69] views low-carbon actions as influenced by multi-level factors like family, community, and policies, showing the importance of external support in fostering sustainable practices. These theories offer valuable insights for studying older adults’ climate behaviors, highlighting influences across individual, social, cultural, and policy dimensions. These frameworks provide a strong basis for developing an interview framework and data analysis.

Older adults face challenges in contributing to climate action, including health constraints, resource limitations, technological barriers, and societal perceptions. However, their experience, influence, and capacity for intergenerational collaboration present significant opportunities. Although these obstacles cannot be overcome immediately, incremental measures can help mitigate them and encourage greater engagement of older adults in climate initiatives.

Including older adults in climate action is both a matter of justice and an essential step in addressing the climate crisis. As direct stakeholders, they ensure their needs are reflected in policies, while their expertise and traditional knowledge provide critical insights for ecological adaptation. Older adults also hold social influence, driving action at both the community and policy levels and fostering intergenerational equity. By involving older people in climate initiatives, we can work toward a more inclusive and sustainable future. This research enhances their representation in climate dialogues, empowering them to voice their needs and contribute meaningfully to climate solutions.

## 3. Methodology

This study involved focus group interviews with older adults residing in the UK, aiming to gather their perspectives on climate change and related practices. Additionally, it explored their motivations for learning about climate change and their suggestions for promoting low-carbon lifestyles.

### 3.1. Recruitment and Procedure

This study took place between August 2023 and January 2024, recruiting 30 older participants across seven focus groups. These older adults were from West London, with some participants recruited through the Brunel University Older People Reference Group. Other participants were recruited from Uxbridge Library and churches in West London. The participation rate for this study was 100%, with all 30 individuals attending the focus groups.

Focus groups are a qualitative research method aimed at gathering in-depth views and opinions on a specific topic [70]. Typically, a focus group consists of a moderator and six or more participants. By combining participant dynamics, body language, and responses to questions [71], this method can provide rich insights into future products, services, and system features [72]. In this study, focus groups created a safe and open environment where older adults could freely express their views, helping to reduce participant anxiety. The discussions enabled participants to inspire one another, leading to new ideas or perspectives. The group setting facilitated dynamic interactions, which are less likely to occur in one-on-one settings. While individual interviews were not conducted in this study, the focus group discussions allowed researchers to observe commonalities and differences among participants, providing a comprehensive understanding of their thoughts and motivations.

### 3.2. Sampling Strategy

A purposive sampling strategy was used to recruit older participants, which is suitable for situations requiring an in-depth understanding of a specific group’s views, experiences, or behaviors [73]. This approach focuses on selecting information-rich cases to provide deeper insights into the phenomenon under study, rather than aiming for statistical representativeness. The recruitment criteria did not require participants to have extensive knowledge of climate change—interest in the topic was sufficient. This ensured a range of perspectives and contributed to data diversity. However, older adults who were uninterested in climate change or the workshops were not included in the study, as they declined the invitation due to reasons such as lack of interest in climate change, unwillingness to share their views or scheduling conflicts.

The main participants were recruited from the Brunel Older People’s Reference Group (BORG). Invitations were sent to BORG members via email, explaining the purpose of the study and inviting them to participate in focus group discussions. BORG members were also encouraged to share these invitations with their networks. Additionally, recruitment efforts extended to Uxbridge Library and churches in West London, where posters were displayed on community boards and informational cards were distributed to older adults visiting these locations. Interested participants were asked to express their willingness to participate by emailing or texting the researchers, allowing the team to schedule sessions at mutually convenient times.

Inclusion criteria for this study involved age and willingness to participate. Participants were screened with the question “Are you willing to discuss climate change?” and needed to be aged between 60 and 85, be able to move independently, and communicate fluently. Those who met the inclusion criteria were invited to the study.

This study aimed to explore older adults’ perceptions and behaviors regarding climate change, focusing on a group capable of engaging in in-depth discussions. This approach sought to provide a detailed understanding of the low-carbon behaviors and climate change perceptions of this specific group, laying the groundwork for future research.

Given that the workshops were held at Brunel University and were highly interactive, participants needed to travel to a designated location and actively contribute to discussions. As such, selecting individuals with mobility and effective communication skills was essential to ensure data quality and facilitate smooth interactions.

The sample size of 30 participants was chosen to ensure a balance between obtaining diverse perspectives and achieving data saturation, where no new themes emerge from additional data collection.

The demographic characteristics of the participants are summarized in Table 1.

### 3.3. Materials Used in Focus Groups

The semi-structured questionnaire used in this study was designed based on established theoretical frameworks and a comprehensive literature review to ensure relevance and comprehensive coverage of the topics.

Prior to conducting the formal focus groups, the semi-structured questionnaire underwent pretesting and validation. The first round of pretesting involved four researchers from Brunel University’s Design School, who provided feedback on the focus group process and materials. This led to improvements, such as offering clearer explanations for technical terms (e.g., “net-zero” or “carbon footprint”) and optimizing the session duration to 120 min, while ensuring refreshments were provided. The second round of pretesting involved three participants from the target demographic in a simulated interview setting. Based on their feedback, adjustments were made to the questionnaire, including increasing font size for readability. Finally, the questionnaire was validated through a pilot test with seven participants, confirming its logical consistency and measurement reliability for use in the main study. Before assigning participants to focus groups, researchers recorded each individual’s level of climate change awareness in an information form. Each focus group, consisting of 4–5 participants, was carefully balanced to include both men and women and participants with varying levels of climate engagement, ensuring a well-rounded and diverse discussion dynamic.

The materials used during the focus groups included printed or digital questionnaires, depending on the participants’ preferences, as well as participant information sheets and informed consent forms. The printed and digital materials contained identical content, with the digital version accessible on participants’ mobile devices. Participants were able to electronically sign the consent forms, which could then be sent directly to the researchers via email. Participants were asked to review the information sheet and provide informed consent before the discussion began. The researcher then introduced the project background and explained the procedure, after which the semi-structured focus groups commenced.

Based on the semi-structured questionnaire, participants engaged in semi-open discussions during the focus groups. Two researchers facilitated the sessions: one managed the flow, asked questions, and ensured all participants had equal opportunities to speak, while the other recorded audio and noted non-verbal cues, stepping in to redirect discussions or address participant discomfort if necessary. Given that climate change topics can evoke concerns about life safety, health threats, or future uncertainties—potentially leading to anxiety, sadness, or anger—the researchers were attentive to emotional signals. They encouraged participants to express their emotions freely without fear of judgment. Questions about financial or housing situations were avoided, with indirect or open-ended questions used instead to prevent feelings of blame or guilt among participants.

The focus group questionnaire covered the following key areas:


**Awareness and Perception of Climate Change**


The role of perception in understanding climate change;Awareness of terms such as “net zero”, “global warming”, and “climate change”.


**Low-Carbon Practices and Motivation**


Experiences with low-carbon practices;Motivations for adopting low-carbon behaviors.


**Access to and Sharing of Climate Change Information**


Preferred channels and methods for accessing climate change information;Strategies for sharing climate change information.


**Expectations and Challenges for the Future**


Personal challenges and aspirations regarding climate change;Expectations of society, government, and businesses in addressing climate change.

Each interview lasted approximately 90–120 min. A 10-min break was provided every 60 min, allowing participants to drink water, move around, or temporarily step away from the discussion. Participants were given no incentive to participate, but tea, water, and light snacks were offered during the breaks to help participants recharge. Additionally, participants were given the option to pause or withdraw from the discussion at any point if they felt fatigued, ensuring their comfort and avoiding any sense of obligation to continue. The interviews from focus group sessions were audio-recorded and transcribed by the interviewer (the first author). To protect participant confidentiality, the data was anonymized.

### 3.4. Data Analysis

The interview data were analyzed using thematic analysis with the support of NVivo software for coding. Data were securely stored on Brunel University’s cloud drive. Transcription and initial coding were carried out by the first author to ensure consistency in data preparation. Following Bryman’s principles of social research, we critically reflected on biases during data collection and analysis and used triangulation methods to enhance dependability and trustworthiness [74]. This study employed Investigator Triangulation and Data Source Triangulation. For investigator triangulation, another researcher reviewed and analyzed the data, followed by team discussions to consolidate themes and minimize potential biases. Data source triangulation was achieved by recruiting participants from diverse settings, including libraries, churches, and older people’s reference groups, ensuring a range of climate knowledge levels and backgrounds.

The data analysis followed Braun and Clarke’s six-phase framework: familiarizing with the data, generating initial codes, searching for themes, discussing and validating themes with co-researchers, defining and naming the themes, and producing the final report [75]. The analysis began with an in-depth review of the transcribed focus group discussions to gain familiarity with participants’ insights and interactions. From this, initial codes were generated to capture recurring ideas and patterns within the data. For instance, “frugality” was coded based on participants frequently referencing resource-saving habits shaped by past economic hardships. Similarly, “intergenerational dialogue” emerged from discussions about sharing climate-related knowledge with younger family members.

Two researchers were involved in the coding process. Initial codes were generated by the first author and reviewed by the third author. Disagreements in coding, such as whether “low trust in government policies” should form a separate theme or be included under “systemic barriers”, were resolved through iterative discussions. This collaborative approach helped ensure that diverse perspectives were considered and potential biases were minimized.

The preliminary findings and themes were shared with older adults during Brunel University’s Older People’s Day event. This engagement allowed participants to provide feedback and validate key themes.

## 4. Results

In this study, the data reveal that older adults perceive and understand climate change. In particular, they show that they describe the impact of climate change on their lives within today’s information-rich society, as well as how they access and interpret climate-related information and knowledge. The participants also discussed their low-carbon practices and the motivations driving these behaviors. Additionally, they shared the challenges they currently face in relation to climate change and their hopes and expectations for the future. Figure 1 is a frame diagram that shows the classification of subthemes and themes of the codes.

### 4.1. Suggestions for Promoting Climate Action


#### 4.1.1. Terminology Old People Can Understand

In the focus groups, the participants explored various terminologies used to describe climate issues, such as “net zero”, “climate change”, and “global warming”. Initially, many older adults were more familiar with the term “global warming”. One participant explained, “We use that very often… When you look at a problem like flooding, we always say it’s related to global warming. People don’t believe it’s global warming, but it is” (Female, 72). However, as some regions experienced shifts that were not exclusively warming, the term “climate change” became more appropriate. As another participant noted, “There are parts of the world where it’s cooling… By describing it as climate change, it’s more inclusive” (Male, 65). This shift from “global warming” to “climate change” resonated with the participants as a more accurate reflection of the broad environmental changes they were witnessing.

While the term “net zero” was recognized as a useful concept for balancing greenhouse gas emissions, it was met with some hesitation. Many participants found the term abstract and felt it lacked clarity. One participant expressed, “It’s a strange term… It seems to be central to government policy… For most people, maybe it’s easier to understand `climate change”’ (Male, 68). This reflects a preference for simpler, more familiar language that feels more grounded in everyday understanding.

Moreover, the participants’ aversion to “net zero” stemmed from their perception that it is primarily a governmental term, one that carries a sense of obligation and pressure. “When you say zero emissions, I think of cars… It feels like trying to fake information”, one participant explained (Male, 68). Another added, “I feel a lot of pressure when you say net zero. I don’t know about net zero” (Female, 81), highlighting how the technical terminology can feel inaccessible or even alienating to this demographic.

To enhance understanding, “net zero” could be explained through relatable examples, such as households using renewable energy to offset emissions from daily activities. This approach ensures that the terminology remains accessible while preserving its conceptual depth.

#### 4.1.2. Cultural Differences and Cultural Integration

In the focus groups, older participants also discussed the role of cultural inclusivity and integration in relation to climate change. The UK is home to immigrants from diverse backgrounds, including many older adults who have come from different parts of the world. One participant emphasized the importance of multicultural environments. Her home was a guesthouse, receiving visitors from different countries, and she talked with everyone. She said, “It really helps if you live in a multicultural house… There will often be enough people in the same situation” (Female, 63). This highlights how cultural exchange provides an opportunity for mutual learning between people from different ethnic backgrounds.

Older adults from developing countries, for instance, often practice low-carbon behaviors due to different economic and social systems. As one Indian participant explained, “In India, we have learned to think about the future… You have to save… Planning for a rainy day and is for emergencies” (Female, 78). This emphasis on future planning and saving contributes to lower household carbon emissions, contrasting with the participant’s perception of Western consumption patterns.

Additionally, some participants expressed the need for immigrant seniors in the UK to better engage with local environmental efforts. One participant noted, “If you don’t understand climate change in the UK… you can’t participate in community activities” (Male, 65). This reflects the challenges immigrant older adults face in integrating into climate-related community activities due to language barriers and cultural isolation.

By sharing these perspectives, this study underscores the importance of cultural inclusivity in climate communication and action, as well as the need to provide immigrant seniors with more accessible information to enhance their participation in environmental initiatives.

#### 4.1.3. Multi-Channel Information Dissemination—By Population Group

In the focus groups, older adults shared their views on reliable sources of information and where they typically obtain knowledge about climate change. The BBC, as a long-established media outlet in the UK, was highly regarded by participants as a trusted source. One participant commented, “The BBC has improved a lot. There were many climate deniers until the BBC spread the word… Public awareness has increased, and you don’t hear counterarguments in the media anymore” (Male, 74). This reflects the role of traditional media in shaping public discourse on climate change.

Newspapers and radio were also integral parts of daily life for many older adults, and they were considered reliable sources of information. One participant explained, “And newspapers, and radio. It can convince the masses. It has accelerated public awareness of climate change” (Female, 62). This demonstrates the importance of familiar, accessible media platforms in informing older generations.

While some older adults have started using computers, many find reading reports on websites tedious and exhausting. Television, particularly for documentaries and news, has remained a trusted source for many. “TV can provide you with some good documentaries, so I believe this is the one” (Female, 75). The reliance on these traditional forms of media highlights the accessibility challenges older adults face when engaging with digital platforms.

In contrast, many older adults expressed distrust in social media, viewing information as unreliable and unsuitable for their needs. As one participant put it, “I’m not a fanatic who needs to search for anything. If there’s something on social media, I don’t read it” (Female, 78). This points to a significant digital divide in how older generations engage with information compared with younger audiences.

Moreover, advice from friends and family was seen as highly valuable. Participants emphasized the importance of intergenerational dialogue. “Certain family members are more interested in climate change… My daughter-in-law is very concerned about it, and my son is very passionate about it” (Female, 70). The quote underscores the significance of trusted personal relationships in shaping views on climate change.

Communities and social groups also emerged as reliable sources of information, with discussions often sparked by interactions with friends and family. One participant mentioned, “I joined a lot of groups… Maybe I will join one on climate change to gain more knowledge and then do more” (Female, 78). This highlights the potential of community-based networks to foster greater climate awareness and action among older adults.

By weaving these quotes into the analysis, we see the complex relationship between older adults and the sources of information they trust, with traditional media and personal networks playing pivotal roles in shaping their understanding of climate change. However, significant gaps remain in the use of digital media among older adults.

### 4.2. Sharing of Low-Carbon Practices

#### Promoting Low-Carbon Practices of the Older People—To Inspire More People

In this study, all 30 participants shared their low-carbon practices, which encompassed areas such as food, housing, transportation, education, and social responsibility. Some participants mentioned that health issues have limited their ability to engage in certain actions, but overall, they demonstrated a range of sustainable behaviors.

Regarding food, the older adults highlighted efforts to reduce food waste. One participant shared, “We try not to throw away food. We respect the people who grow our food, and that’s important. I compost the vegetable peels and return them to the soil” (Male, 79). This illustrates a common practice of composting to reduce waste and grow plants, reflecting a respect for the food production process. Participants were also aware of the higher carbon emissions associated with red meat compared with vegetables, with many reducing their consumption of meat as they age due to both health concerns and environmental awareness. As one participant noted, “They should cut down on meat; it takes a lot of the world to break down meat. We need to talk about consumption” (Female, 62).

In terms of transportation, many participants mentioned that they had reduced their use of private cars and increasingly relied on public transport, particularly in London, where it is readily available. One participant stated, “I live in London, so public transport is very convenient for me. I always take the bus or the subway” (Female, 65). Mobility challenges due to age were acknowledged, but older adults continued to prioritize public transportation whenever possible. Many participants had also reduced long-distance travel, a trend that was accelerated by the COVID-19 pandemic. “During the last three years of the pandemic, I have reduced my travel,” one participant shared (Male, 74). Changes in family structure, such as the loss of a spouse, also contributed to a reduction in long-distance travel among older adults living alone.

Recycling was another area where participants shared practical knowledge. Older adults discussed collecting rainwater for plants, composting, donating old clothes, and turning off lights to conserve energy. They also emphasized the importance of recycling plastics due to the environmental harm caused by their slow degradation. One participant explained, “When I buy goods, I reuse plastic bottles rather than throw them away after one use. Buying refill packs is more cost-effective and eco-friendlier” (Female, 63).

Energy-saving practices during winter were particularly significant due to rising energy costs. Many older adults reported using products like electric blankets or hot water bottles to reduce heating bills while staying warm. Others mentioned turning off heating at appropriate times or lowering the thermostat to save energy. “I turn down the heat. I go around calculating my electricity and gas bills,” one participant shared (Female, 69). Additionally, some participants had installed insulation systems or smart meters to improve energy efficiency and further reduce costs. The energy crisis has prompted some participants to realize that energy use needs to pay a higher price, thus triggering energy-saving behavior. However, some participants said they have already developed energy-saving habits.

Several participants attributed their low-carbon behaviors to values of frugality in-stilled during their upbringing, particularly those who had lived through the post-World War II era or periods of food rationing. “We were very frugal. We saved things, and my friends are still the same today,” one participant recalled (Male, 84). These experiences shaped a long-term approach to resource use and environmental responsibility. For others, a sense of social responsibility motivated their actions. One participant emphasized the broader perspective of climate action, stating, “Reusing clothes is important for sustainability” (Male, 68).

These low-carbon behaviors, such as reducing food waste, using public transportation, and recycling, show that older adults are actively engaged in sustainable practices. These actions are often guided by personal values like frugality and supported by policies and social structures.

### 4.3. Strategies for Exploring Climate Change

#### 4.3.1. Discuss Climate Change—History, Geography, and Perception

It is noteworthy that, as long-term residents of the UK, older adults shared their perceptions of climate change by recounting changes in temperature, smells, and air quality. Their reflections were deeply rooted in personal experiences and memories from various periods, almost as if they were assembling scenes from past films. These vivid recollections not only enhanced the discussion but also evoked emotional resonance, inviting others to relate to their own memories.

One participant recalled, “I remember the heatwave in 1976, the first one. The weather was beautiful. We went boating on the sea between Ireland. I remember dipping my toes in the water, and it was freezing, absolutely freezing. I couldn’t believe it—so cold, and yet, there were turtles swimming around” (Male, 77).

Older adults described their understanding of climate change from a geographical perspective. While climate change is a global issue, they were particularly attuned to the changes in their local environments. In discussing such a vast topic, personal experiences often resonate more strongly, particularly when these experiences are tied to visible environmental shifts. These specific and localized details make climate change feel less abstract and more tangible, emphasizing its direct impact on individual lives.

Older adults often draw on personal historical experiences to understand climate change, linking it to specific events such as heat waves or changes in air quality. This localized way of understanding makes climate change more tangible and relatable.

#### 4.3.2. Awaken Awareness of the Natural Environment—Land, Sea, Plants, Disasters, Air, and Animals

The older participants also discussed nature’s warnings to humanity, citing phenomena such as glacier melting, deforestation, wildfires, and heat waves. Their concerns went beyond the increasing frequency and intensity of natural disasters, focusing on the long-term implications these changes may have on the survival of both humans and animals. Whether the information came from friends or television news, the older adults vividly recalled numerous details, expressing profound regret and worry about the current state of the environment.

“It wasn’t until April of this year that people started to really worry about the 5 °C rise in ocean temperatures in Ireland. In Antarctica, we’ve reached the limit of how much carbon dioxide the ocean can absorb, and that’s why we’re having these massive heat waves. The snow is melting, and polar bears have nowhere else to go” (Female, 69).

Older adults bring personal reflections, memories, and concerns about the future into discussions on climate change. This personal approach not only enriches the dialogue but also adds emotional depth. Their lived experiences serve as powerful evidence in the climate change conversation, demanding greater attention and reflection on this global issue.

#### 4.3.3. For the Next Generation—The Motivation

Older adults also discussed their motivations for caring about climate change, with a common concern being the well-being of their children and grandchildren. The most frequently cited motivation was the desire to leave a better environment for future generations. Participants emphasized that this concern is likely shared across the older population, as most older adults, regardless of their background, care deeply about the quality of life for the next generation.

“You want your children and grandchildren to have a truly livable earth too. I think that’s true for everyone, no matter where they come from. If they have a family, if they have grandchildren, it’s a shared motivation in the older community” (Female, 82).

“We talk about it all the time—for the future of the planet, for the children’s children. When you worry about the next generation or consider it a responsibility, you feel you have something to contribute” (Female, 63).

Leveraging older adults’ motivation to safeguard the future of their children and grandchildren can be an extremely effective strategy for promoting climate change action. This motivation transcends cultural, national, and background differences, making it a powerful and universally resonant approach.

#### 4.3.4. Attracting Attention with Economic Factors

Economic factors emerged as a critical concern for older adults, serving as a primary motivator for adopting low-carbon practices. The drive to “save money” was frequently mentioned as a reason for exploring new energy technologies. As one participant remarked, “Everyone is motivated to save money, which is why things like insulating your house actually save money in the long run. This will not only reduce the impact on the climate but also save money“ (Female, 82). This highlights how financial considerations intertwine with environmental consciousness, driving low-carbon practices among older individuals. In practical terms, many participants reported reducing their heating usage or adopting cost-saving technologies like electric blankets to stay warm during the winter. This demonstrates a practical response to the increasing costs of energy. Moreover, participants discussed how a system of rewards and penalties could further encourage such behavior. One participant highlighted the importance of financial incentives for switching to new energy vehicles. ”They reward owners with a 100 grant and the ability to rent an electric car for a month. I think if they give people more rewards… it would encourage people“ (Male, 65). This suggests that financial incentives could play a significant role in fostering environmentally friendly behavior, particularly when they are directly linked to tangible benefits like cost savings. Conversely, participants also mentioned how penalties related to older vehicles, such as additional fees, act as a deterrent and nudge towards sustainable alternatives. As one participant noted, ”Now if you have an old car, you have to pay the fee every day. It’s all about protecting the environment and making people aware of it“ (Male, 68). These statements reflect a broader understanding among older adults of how financial incentives and penalties can regulate environmental behavior.

However, several participants expressed frustration over the high costs of implementing energy-efficient measures such as solar panels, which they felt were prohibitive. One participant lamented, “Some of our houses don’t have solar panels… The cost of installing solar panels is high. At our age, it’s just too expensive” (Female, 82). Another added, ”I did the maths. If I fitted the house with solar panels, it would cost about GBP 15,000. I might die before I make any money. It’s too expensive“ (Female, 78). This reveals a significant barrier to the adoption of low-carbon technologies, particularly among older adults who may have financial limitations and shorter time horizons for recouping the investment. When encouraging older adults to adopt low-carbon practices, economic factors are clearly a crucial consideration. Saving money is another key motivator for this group, and both rewards and penalties are viewed as effective mechanisms for regulating behavior. However, the financial realities faced by older people—such as the high costs of home renovations or electric vehicles—mean that ”saving money” alone may not be enough to address the challenges. Therefore, in designing low-carbon policies and initiatives, it is essential to take older adults’ economic circumstances into account and offer more flexible and affordable options.

In addition to economic incentives, emotional appeals, and social influences can significantly motivate behavior change. For instance, emphasizing the emotional satisfaction of contributing to a better future for grandchildren or showcasing community leaders engaging in low-carbon practices can inspire collective action.

#### 4.3.5. Provide Guidelines for Learners at Different Stages

To address the varying levels of understanding and engagement with climate change among older adults, it is crucial to acknowledge their diverse motivations and needs. Some participants in this study were at the early stages of awareness, showing a desire for more information and guidance to take meaningful action. While certain individuals have already adopted low-carbon practices, they often expressed a need for additional resources to continue making progress. This reflects a broader challenge in climate change communication—ensuring that those who are already taking steps toward sustainability have access to the necessary knowledge and support to do more.

However, for some older adults, age plays a significant role in their level of motivation. As one participant noted, their advanced age made them feel less inclined to act, as they may not live long enough to witness the long-term effects of climate change. “I don’t have the motivation to do it anymore. I’m very old” (Female, 78). This sentiment highlights a potential barrier for engaging older populations in climate action, as personal relevance and future-oriented concerns may be diminished with age.

Skepticism about climate change was also evident among the participants. Some voiced doubts about current sustainability practices, particularly with regard to the integrity of waste management systems. For instance, one participant questioned the credibility of European countries’ approach to waste management, particularly the practice of exporting plastic waste. “Have you heard the one about European countries exporting their plastic waste to other places? How ridiculous do you think that is?” (Male, 60). This critique reflects concerns about the ethical implications of waste disposal and the perception of ineffective global solutions to environmental issues.

Additionally, skepticism extended to the broader sustainability movement, with doubts about individual actions, such as dietary changes, making a real impact on the climate crisis. These reflections suggest that some older adults are not fully convinced of the effectiveness of widely promoted sustainability efforts, particularly when such efforts seem disconnected from broader, systemic environmental challenges.

By recognizing the diverse attitudes toward climate change—ranging from motivation to skepticism—this study underscores the need for tailored communication and engagement strategies. To address apathy and cognitive dissonance, communication strategies should emphasize personal relevance by linking climate change to local environmental impacts, such as rising energy costs or extreme weather events. Highlighting collective benefits can also foster a sense of agency and reduce feelings of helplessness.

#### 4.3.6. How the Message Is Presented—Remind, Warn, Encourage, and Attract

In the focus groups, participants discussed various methods to remind people about the impact of their actions and motivate them to engage in more environmentally responsible behavior. Older adults frequently mentioned reward and penalty systems as formal mechanisms that can regulate behavior. These systems not only influence individuals through incentives and sanctions but also provide information that evaluates specific actions. Such information serves as a subtle reminder to reflect on whether their behavior aligns with environmental standards. For example, older adults referred to news reports showing images of the destruction of the Amazon rain forest or homeless animals, which served as “soft nudges”, helping them realize the environmental consequences of their daily choices. One participant remarked, “When I see those pictures of animals losing their habitats, it makes me think—what have I done today that could be hurting the planet? ” (Female, 68). Rather than delivering an emotionally charged message, these reminders gently prompt individuals to reconsider their behavior, fostering self-awareness in a non-intrusive way.

Beyond reminders, participants highlighted the importance of warnings that emphasize the potential consequences of inaction. Warnings, compared with gentle nudges, deliver a more serious message aimed at capturing attention and informing individuals of the risks associated with their current behavior. These warnings can evoke stronger emotional responses and create a sense of urgency or responsibility. For instance, one participant stated, “If you tell people about fines for leaving the heating on too high, they’ll listen. But just saying it’s about saving the planet? Not everyone cares as much” (Male, 72). This reflects the perception that monetary penalties may be more effective than moral appeals in motivating behavioral changes.

In contrast to warnings, rewards offer positive reinforcement. Older adults expressed interest in financial incentives, such as subsidies for switching to renewable energy sources. These tangible rewards could help encourage environmentally conscious behavior. However, rewards need not always be financial. Verbal encouragement and positive reinforcement, such as acknowledging good practices or providing feedback through informational cues, can serve as powerful motivators. One participant shared, “Even if it’s just a thank-you letter for participating in a recycling program, I’d feel good about it. It makes you want to do more” (Female, 74). Another noted, “When someone says, ‘You’ve saved energy this month,’ it feels like you’ve achieved something. That’s encouragement enough” (Male, 65). Encouragement builds a supportive environment by praising positive actions, which helps sustain long-term engagement with low-carbon practices.

Finally, participants discussed the role of attraction as a motivational strategy, with the goal of drawing attention through visually engaging or creatively designed content. Rather than directly prompting action, attraction works by sparking curiosity and stimulating exploration, encouraging individuals to learn more about climate change. One participant noted, “I love those colorful infographics that show you how small actions add up to big changes—they’re so much easier to understand than just numbers” (Female, 71). Another added, “If it’s a documentary with a good story, I’ll watch it. But if it’s too technical, I lose interest” (Male, 67). By engaging people through appealing visuals or compelling stories, attraction can lead to greater awareness and, ultimately, more sustained involvement in climate action.

#### 4.3.7. The Importance of Visual Materials

Simplifying information and presenting it in a format that is easily understandable for older adults is essential for effective communication on climate change. One participant explained, “I think whatever channel of information you want to get the message across, you need to make the message very easy to understand, and that’s one of the reasons I think climate change is better than net zero. All these people must understand what is necessary… It doesn’t matter which channel you use. What matters is that it doesn’t work. Otherwise, people won’t be motivated to download a lengthy, boring report” (Female, 82). This emphasizes the need for straightforward language and strong visual elements when communicating climate issues to older adults.

Visual materials, such as images and videos, are often more effective in conveying messages, particularly for older individuals who may experience vision impairments or difficulties with reading. Visual communication has the power to evoke empathy and emotional resonance, making it a useful tool for raising awareness about climate change. One participant mentioned, “Perhaps visual material is easier to understand” (Female, 70), while another echoed this sentiment, stating, “The pictures are good. This is universal” (Female, 69).

Participants also stressed the importance of keeping the message simple and accessible. As one older adult advised, “Keep it simple, silly. Have you ever heard that phrase? Keep it simple, stupid, and don’t be stupid enough to make things unnecessarily complicated. Keep it simple. I like simple information” (Male, 70). This reinforces the idea that straightforward, visually engaging communication is likely to be more effective in motivating older adults to engage with climate-related content.

### 4.4. The Need for Climate Action

#### 4.4.1. The Need for Government Improvements

Older adults, having lived in the UK for many years or even generations, are particularly attuned to shifts in local government policies, often noticing contradictions in government climate initiatives. They pointed out inconsistencies, such as allowing polluting businesses to operate while simultaneously promoting climate awareness. One participant noted the irony of climate change demonstrations turning violent, undermining the very cause they were advocating. Additionally, a retired town hall employee highlighted flaws in the recycling system, observing that waste sent to main recycling stations was often not properly processed. “It’s really not separated… Not all of it gets recycled” (Female, 63). This reflects broader concerns among older adults about the gaps between government policies and their implementation. Many older participants called on the government to refine its messaging and management strategies, suggesting that current efforts lack coherence and effectiveness. They expressed a need for the government to be more consistent and transparent in its approach to promoting and executing climate change initiatives. As one participant explained, “I think the government has to invest. The government has to do something. It helps us tell the people. Hopefully they will accept and learn” (Female, 81). Another voiced concerns about the integrity of political leaders, stating, “People are very worried about the honesty of the politicians who lead us. We are concerned about the honesty of the information we are getting from British politicians” (Male, 62).

The older adults also highlighted a lack of transparency and openness in how the government implements and reports on climate policies. This has left many feeling frustrated, with limited channels for communicating their concerns. Furthermore, the absence of a clear system of rewards and penalties for low-carbon actions has resulted in a lack of public motivation to engage in environmentally friendly practices. This underscores the need for more structured incentives and improved communication to foster greater participation in climate action. While participants criticized government inconsistencies, they also recognized the potential for collaboration. For example, establishing local advisory panels with older adults or hosting public consultations could enhance policy transparency and inclusivity.

Beyond individual efforts, systemic change is necessary to address structural barriers to sustainability. Policies that support renewable energy transitions, such as community-level solar projects or subsidies for electric vehicles, create enabling environments for low-carbon lifestyles. Additionally, businesses can play a pivotal role by adopting circular economy principles, which reduce waste and encourage sustainable consumption patterns.

#### 4.4.2. Sustainability Awareness in Business

Older adults emphasized that addressing climate change is a collective responsibility, extending beyond individuals to include various industries and businesses. They suggested that shifting from non-sustainable to sustainable consumption patterns can be effectively encouraged through innovative marketing strategies. For example, when products clearly communicate their sustainability credentials, consumers are more likely to engage in sustainable behaviors through their purchasing decisions. This indirect approach can be highly effective in fostering environmentally conscious habits. As one participant noted, “Reusing clothes is important for sustainability. So one area I talked about was branding. Brands like Patagonia… they encourage people to mend clothes and not throw them away. They gave away repair tools. So there are sustainable behaviors in some brands” (Male, 65).

Some participants also highlighted the potential of sustainable business models, such as incentivizing the recycling of plastic bottles. They suggested that such initiatives could be driven by creative marketing strategies, which align economic incentives with environmental goals. “Some people say that if you pay people to recycle plastic bottles, you can look at who makes this marketing model and make money in this way” (Female, 72). This perspective underscores the role businesses can play in shaping consumer behavior towards more sustainable practices, aligning individual actions with broader environmental objectives through subtle yet impactful marketing tactics.

#### 4.4.3. Improvements in Industry Support Facilities

In the focus group discussions, older participants highlighted specific challenges regarding the promotion of electric vehicles, particularly the lack of adequate infrastructure. While the UK has made significant efforts to promote electric vehicles, participants noted that supporting infrastructure, such as charging stations, remains underdeveloped, making it difficult for them to transition to electric vehicles. One participant emphasized the perceived inconvenience, stating, “I don’t think I want an electric car. It’s too much trouble. It needs to be recharged” (Male, 60). This concern reflects the broader need for more accessible and user-friendly infrastructure to support electric vehicle adoption, especially for older individuals who may be less familiar with new technologies.

Additionally, participants raised concerns about the challenges that public transport presents, particularly due to age-related health conditions. Long distances and the physical demands of using public transport were frequently mentioned as barriers. One participant reflected on their daily efforts to stay active but noted, “Walking is difficult in winter. The cold weather made walking a great challenge” (Female, 78). Another participant pointed out the importance of living near transport facilities, stating, “As you get older, transportation gets harder, and it’s really important to live near a bus stop or something” (Male, 84). These comments underline the importance of incorporating elderly-friendly designs into public transport systems and improving infrastructure to meet the needs of an aging population.

## 5. Discussion

This study sheds light on the important role older adults can play in climate action. It highlights their potential as active contributors and advocates for sustainable practices, rather than passive recipients of climate policies. This research explores their views and behaviors toward climate change, providing valuable insights into their motivations and the challenges they face.

**Theme 1** shows that older adults prefer familiar terms like “climate change” over abstract ideas such as “net zero”. This finding emphasizes the need for clear and simple language to encourage participation. This study also highlights the role of cultural integration, which helps share knowledge and influence behaviors. Participants from developing countries demonstrated strong habits of saving energy, showing the value of cross-cultural experiences in shaping global climate strategies. Traditional media like the BBC, newspapers, and television remain key sources of information for older adults. In contrast, digital platforms and social media are used less often, partly due to trust issues. Family and friends play an important role in shaping their understanding of climate issues, showing the value of intergenerational conversations in driving climate action.

**Theme 2** highlights established low-carbon habits among older adults, such as reducing food waste, using public transport, and eating less meat. These actions are driven by both cost savings and a strong sense of responsibility. However, financial barriers, such as the high cost of solar panels or electric vehicles, limit their ability to adopt more sustainable technologies. Policymakers should address these barriers by offering more affordable and flexible solutions.

**Theme 3** reveals how older adults make climate discussions more relatable by sharing personal stories and emotional reflections. Their lived experiences help others connect to the real impacts of climate change. Nature is another factor that motivates people to take action, and the plight of animals and plants can gain people’s sympathy. Moreover, participants mentioned the motivation of protecting the earth for the next generation. Intergenerational dialogue also emerged as a powerful tool, with older adults sharing stories and practical advice with family members.

To improve climate communication, this study suggests simplifying information, using visuals, and tailoring messages to older adults. These strategies include practical reminders for daily actions, warnings about risks, and sharing success stories to inspire change.

In **Theme 4,** participants expressed frustration with unclear government policies, calling for more transparency in reward and penalty systems. They also emphasized the need for businesses to take a more active role in promoting sustainability by driving eco-friendly behaviors from within.

Current literature predominantly addresses the health and psychological impacts of climate change on older adults, emphasizing their vulnerability to climate-related risks, such as heat-induced illnesses during heatwaves [24], due to limited adaptive capacities stemming from physical, economic, and social constraints. Despite varying levels of climate change awareness, older adults’ participation in climate action, especially in policy development and implementation, remains minimal [76].

This study also uses theoretical guidance to find that multiple theoretical frameworks (biopsychosocial model, social cognitive theory, value-belief-norm theory, and social-ecological systems theory) can be presented from the aspects of individuals, society, culture, and policy. The main findings include the following:**Individual level:** Consistent with the biopsychosocial model, physical health significantly influences older adults’ capacity for low-carbon behaviors. Participants with better mobility engaged more readily in sustainable practices, while those with physical limitations faced greater challenges. Self-efficacy, as highlighted by social cognitive theory, also plays a vital role; older adults with confidence in using green technologies were more likely to adopt low-carbon lifestyles, underscoring the need for targeted interventions to enhance their technological confidence.**Social level:** Aligned with social-ecological systems theory, this study demonstrates the influence of community support on climate engagement. Local environmental initiatives and intergenerational dialogue were pivotal, with participants reporting knowledge exchange with younger family members, strengthening both climate awareness and intergenerational bonds.**Cultural level:** Value-belief-norm theory explains the role of cultural values, such as frugality and resource conservation, in shaping sustainable behaviors. Immigrant participants enriched discussions by introducing diverse low-carbon practices from their home countries, highlighting the potential of cross-cultural exchange for global climate strategies.**Policy level:** Rational choice theory underscores the importance of economic incentives and inclusive policies. While participants acknowledged the role of reward and penalty systems in encouraging sustainable behaviors, they also pointed out significant barriers, such as the high costs of green technologies like solar panels. Dissatisfaction with policy transparency and coherence further emphasized the need for systemic support tailored to older adults’ financial constraints.

There will be further opportunities to strengthen older adults’ involvement in climate change and fill gaps in existing research.

**Broadening research scope:** Future studies should expand to include socioeconomically and culturally diverse samples of older adults. This would allow for a deeper understanding of how perceptions and behaviors toward climate action vary across different geographic, cultural, and economic contexts. For instance, comparative research could explore the unique low-carbon practices adopted by older adults in developing countries, whose resource-conscious lifestyles and traditional knowledge could inform more adaptive climate strategies [77]. Addressing these gaps would enhance the representativeness of research and provide actionable insights for global and local climate policies.

**Economic incentives and accessibility:** Older adults face financial barriers to adopting low-carbon technologies, such as solar panels and electric vehicles. Future research should investigate which economic incentives—such as subsidies, tax rebates, or utility discounts—are most effective in encouraging sustainable practices among this demographic [78]. Additionally, addressing cost barriers associated with green technologies could reveal pathways to make these options more accessible for individuals on fixed incomes. Tailored solutions should focus on reducing upfront costs for individuals on fixed incomes, ensuring that financial constraints do not limit their participation in climate action. These incentives could also be paired with educational initiatives to help older adults better understand and adopt green technologies.

**Intergenerational dialogue:** Older adults play a critical role as climate communicators within their families and communities. Their lived experiences and personal narratives can inspire younger generations to take meaningful action, while younger individuals often provide insights into new technologies and energy solutions. This bidirectional exchange creates opportunities for mutual learning and collective action [60]. Future studies could explore how to empower older adults as advocates for sustainable practices, equipping them with the tools and platforms to effectively share their knowledge and experiences.

**Tailored communication strategies:** Given older adults’ reliance on traditional media, climate communication strategies should integrate traditional and digital platforms to maximize outreach. Simplified, visually engaging materials—such as infographics for beginners or interactive guides for more experienced individuals—can address diverse levels of climate knowledge. Additionally, audio formats like podcasts or narrated videos can cater to a wide range of learning preferences. Hybrid platforms that combine the accessibility of traditional media with the interactivity of digital tools would enable more inclusive and effective engagement with older populations.

**Community integration:** Governments and community organizations should actively involve older adults in local climate initiatives. Hosting interactive events, such as exhibitions or community sustainability days, can create opportunities for peer-to-peer knowledge sharing and intergenerational collaboration. These activities foster a sense of belonging and encourage collective participation in climate action. Furthermore, integrating climate-focused programs into older adults’ daily routines can help normalize sustainable behaviors and enhance their contribution to community-level efforts.

**Systemic changes and cultural integration:** To create a supportive environment for older adults’ climate engagement, systemic changes are essential. Investments in green infrastructure, incentives for sustainable business practices, and community-level programs can enable older adults to adopt low-carbon lifestyles more easily. Linking individual actions, such as conserving energy or reducing waste, to broader societal benefits can help older adults recognize their contributions as part of a collective solution to climate change. This sense of shared purpose inspires ongoing participation and highlights the cumulative impact of small, everyday actions.

**Cultural integration:** It also offers opportunities to promote knowledge sharing and behavioral influence across diverse racial and cultural groups. By providing tailored resources, such as language support and culturally relevant information, older adults from migrant backgrounds can engage more effectively in climate initiatives. Cultural integration fosters creativity and adaptability in climate strategies by combining knowledge and practices from different backgrounds, ultimately enhancing collective climate resilience.

Figure 2 illustrates this study’s climate-related opportunity points across dimensions, such as intergenerational, technological, economic, and societal aspects. Horizontally, the framework spans personal, community, business, government, and global levels, highlighting the intersections where opportunities for engagement emerge. While this study primarily focuses on societal contributions, it identifies future research avenues in intergenerational and global climate communication and the role of emerging technologies.

### Limitations

This study has certain limitations that should be acknowledged. First, the sample size was limited to 30 participants, all residing in West London. While this provided valuable perspectives and rich data, the findings may not fully capture the diversity of older adults’ views and behaviors across different regions or socioeconomic contexts. Expanding the sample size and including participants from a broader geographic range would enhance the representativeness of the aging population.

Additionally, this study focused on older adults who were able to attend group discussions, which may have excluded those aged 85 and above or individuals with mobility or language barriers. Future research could adopt tailored methods, such as one-on-one interviews or flexible, home-based approaches, to better address the needs of these under-represented groups. Such efforts would enable a more comprehensive understanding of the diverse experiences, challenges, and contributions of older adults in relation to climate action.

## 6. Conclusions

This study highlights the unique perspectives of older adults on climate change, emphasizing the importance of accessible language and cultural considerations in promoting low-carbon behaviors. Older adults enrich climate discussions through personal reflections and life experiences, adopting sustainable practices driven by economic savings and a sense of social responsibility. However, the high initial cost of green technologies remains a significant barrier, underscoring the need for more economically viable solutions.

Traditional media and family networks play a crucial role in shaping older adults’ climate awareness, while simplified and visualized content is essential for fostering effective engagement. This study also reveals the need for tailored strategies to address varying levels of understanding among older adults. Dissatisfaction with the transparency of government climate policies further highlights the necessity of clear incentives and accountability mechanisms.

This paper is original in representing the voice of an under-represented group (i.e., older people) in the discourse of climate change. It adopted a rigorous qualitative approach to collect and analyze rich data from 30 older people in West London, and utilized various theories to explain the findings. The study has led to concrete insights into how to incorporate older people’s concerns and motivations in future climate change communication and action.

Future research should explore more diverse cultural contexts, design customized incentive schemes to encourage broader adoption of low-carbon practices and investigate how intergenerational collaboration can enhance older adults’ role in climate action. 

## Figures and Tables

**Figure 1 ijerph-22-00154-f001:**
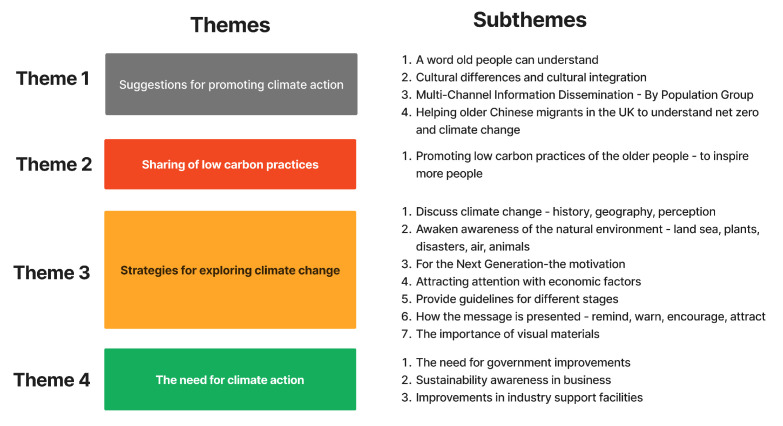
Coded theme classification.

**Figure 2 ijerph-22-00154-f002:**
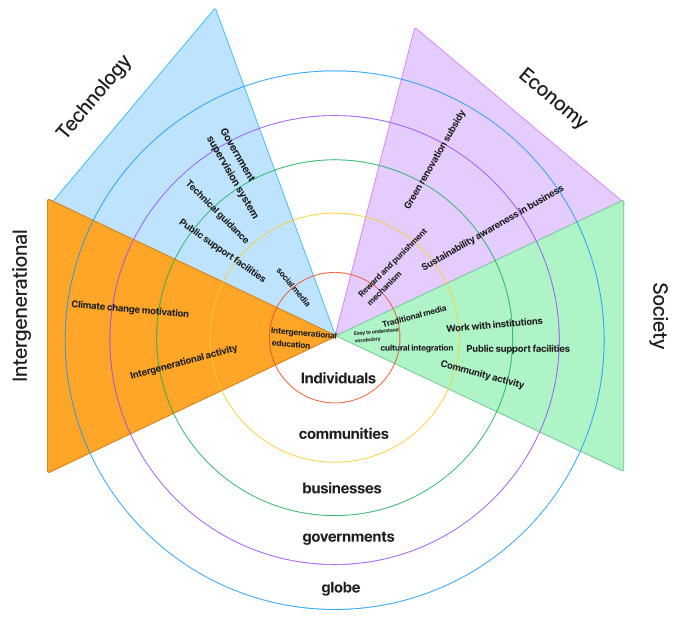
Climate-related opportunity points across dimensions.

**Table 1 ijerph-22-00154-t001:** Demographic characteristics of participants (N = 30 participants).

Category	Sample (%)
Female	60
Male	40
**Age**	
60–70	33
71–80	53
81–85	13
**Race**	
White	83
Asian	13
Black	3
**Climate-related knowledge**	
No knowledge	3
Basic knowledge	40
Moderate knowledge	46
Advanced knowledge	10

## Data Availability

The data relating to this study can be obtained from the first author.

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
