# Peer review of "Investigating Older Adults’ Response to Climate Change"

_ijerph, 2025, doi:10.3390/ijerph22020154_

Round 1
Reviewer 1 Report
Comments and Suggestions for Authors
Thank you for the opportunity to review this paper. This is an interesting and timely topic of discussion. The manuscript, however, needs more work. I have some overall comments and then detailed comments after that. I hope that the authors will find these helpful as they revise.
OVERALL COMMENTS
The paper lacks a theoretical lens with which to approach the topic. Consequently, the paper lacks the rigor of an academic discussion. The authors also need to engage more with existing literature on this topic.
The results are unnecessarily divided into 7 themes. There is a lot of repetition and overlap in the results section. I reckon the results could be “tightened” to 3-4 themes and the discussion points that are currently included with the results could be moved to the discussion section. If necessary, more quotes could be included in the results section.
Since the discussion points are intertwined with the results, the actual discussion section is simply a summary of the results and a very long list of limitations. Importantly, qualitative research is not a limitation. It does not require “quantitative verification” to be considered worthy.
DETAILED COMMENTS
The first sentence should have a couple more citations as the authors state the availability of a “considerable body of literature”.
The first sentence and the rest of the paragraph seem unrelated. The former is about the effects of CC on older adults while the latter refers to causes of/contributions to CC.
Page 1, paragraph 1, it might be useful to acknowledge that the mostly youth-led global climate movement since 2018 has also (often unfairly) blamed older adults for the current climate crisis. This has shaped people’s perceptions of the role of older generations in climate change.
Page 1, line 27: would it be possible to briefly state why this is the case?
In page 1, the term “climate change” has been used to mean climate change cause, impact, as well as climate action. This gets a little confusing for the reader as the content keeps alternating between these. For example, in the first sentence of paragraph 3 (line 28), it is not clear whether the authors refer to CC cause, impact or action when they say CC is influenced by several factors. If they mean CC cause, then the following sentence sounds ageist because it seems to imply that population aging in urban areas is one of the factors causing CC. This is then contradicted by the next sentence that talks about CC impacts on vulnerable groups like older adults. So, a little clarity is needed.
Page 2, line 45: “…contribution to climate change” seems like contribution to CC cause. But the authors actually mean climate action, including education, awareness, resilience, mitigation, and adaptation.
Page 2, line 63: It would be good to begin the sentence with an acknowledgement like, “Although the majority of carbon emissions are driven by fossil fuel consumption and the activities of large corporations, individual behavior can also play a crucial role in addressing the crisis.” Not necessarily this exact sentence, but an acknowledgement of the reality that individual behavior changes, although impactful, are still limited in how much they can achieve in combatting CC. Moreover, low-carbon, sustainable lifestyles may be expensive for some people, at least initially, which can make it difficult for them to make the switch. In other instances, it may not be feasible, such as using EVs in cold climates. So, individual action, to some extent, is limited, whereas a lot more needs to be done by the big polluters.
Page 2, line 86: the term “elderly” is considered ageist. Please replace with older persons/older adults/older people.
Page 3, line 113: it is also true that many older adults are alarmed by the climate crisis and are working/volunteering/advocating tirelessly to be able to leave behind a healthy planet for their grandchildren and future generations. Organisations like Climate Elders Actions, The Raging Grannies International etc. are working in this area.
Page 3, last paragraph: Older adults are usually more likely to vote compared to other age groups and can therefore exert significant influence over a country’s climate policies. Moreover, many countries have older leaders who make climate-related decisions, so it is not entirely accurate to say older adults are not represented.
Page 4, line 167: How did the researchers decide on 30 participants? How were the other participants recruited (those not part of the Brunel University Older Adults Reference Group)? If there were people of different races, it would be good to include that in the table and the text. What about education? There is correlation between educational attainment and climate-related knowledge so it would be good to know the participants’ educational attainment as well. Similarly, marital status and living arrangements can also shape older adults’ engagement in social and community activities, including climate action.
Page 5, line 209: There were individual interviews as well? This is not mentioned before.
Where were the FGDs held? Were the participants responsible for reaching the venue themselves? Did the participants receive any incentive for participation?
Page 5: Data Analysis
It is not enough to merely list the steps of data analysis. Please demonstrate with an example or two how some codes were generated, was there any disconfirming evidence, how many researchers coded the data, were there any conflicts between the researchers while coding, and if so, how were they resolved, what about trustworthiness of data? Did the researchers share their findings with the participants for their feedback? There should also be a citation for thematic analysis.
Page 6, line 228: It might be better to say, “Suggestions for promoting climate action”.
Data Analysis:
Does FG04,03 mean participant number 3 from FGD number 4? If so, this is not immediately clear. It might be better to give the age and sex of the participant instead of serial number because that would help the reader to see how the different age groups and genders are thinking about the topic.
Page 6, line 258: I couldn’t understand this quote in the context mentioned.
Page 8, lines 345-350: Can this be attributed to climate action or is this purely driven by the cost of energy? If heating was affordable, would the participants still choose to turn it off due to climate concerns?
Page 8, theme 4.2: it is not clear how the actions of the participants are inspiring other people. Did the participants mention something to this effect?
Page 12, theme 4.4: this sounds like a repetition of “4.1.1. Terminology old people can understand”
Rewards/penalties/incentives are repeated several times in different themes.
Page 14, lines 657-658: why is qualitative research a limitation?? And why does qualitative research need quantitative verification to have value??
Reviewer 2 Report
Comments and Suggestions for Authors
Incorporate the suggested changes and resubmit it again

Round 2
Reviewer 1 Report
Comments and Suggestions for Authors
Thank you for taking my feedback into account to revise the manuscript. It is a lot clearer and more organized, especially the results section. I have some (mostly) minor comments listed below. However, the methods section needs more work. It might help to have a qualitative researcher look at this section to ensure that the research terms, research processes etc. are described in qualitative language.
Detailed comments
Introduction: it would be good to mention somewhere that this study was undertaken in the UK with London-based older adults—maybe in the paragraph beginning line 68.
Page 3, line 120: citation needed.
Page 3, lines 120-128: I was unable to make the connection between pro-social behaviour and climate action, especially in light of the cited article on socio-emotional selectivity theory. Could the authors add a sentence to link these? Maybe link it to generativity or legacy to show that concern for future generations may lead older adults to engage in pro-social behaviours such as climate action.
Page 4, line 149: citation needed for UNDESA
Page 4, line 152: citation needed for Age UK
Page 4, line 154: citation needed for Green Homes Grant
Page 4, line 165: citations needed for “several studies”
Page 4, paragraph beginning line 166: since there is a separate section for “transport”, the information about dispersed geographical layouts requiring personal transport can go in there. Currently, it is repeated in both points 1 and 2.
Page 4, line 184: It is not clear how “lifestyle” is separate from the other points mentioned in the paragraph. Maybe give an example.
Page 5, line 204: the authors have stated “life cycle perspective” but cited “life course perspective”. These are two different things. Life course perspective is more appropriate in the context of this discussion.
Page 5, line 213: I am not sure it is accurate to say that older adults have lived through similar social and environmental changes as both climate change and technological change at current levels are unprecedented. Maybe replace “similar” with “varied” to cover the changes different cohorts of older adults have seen in their lives?
Page 5, line 232: “senior” is also considered ageist when used to refer to an older person. It is okay in the context of rank and hierarchy (which could certainly apply to older politicians), but maybe replace “senior” with “older”.
Page 5, line 239: could the authors reframe this sentence to exclude “should”? Things that should be done usually go into recommendations at the end of the paper.
Page 6, line 248: Here as well, this is a recommendation that does not fit in here.
Page 6, line 263: Same here. These should go at the end of the paper or in the discussion.
Page 7, line 325: why study 1?
Page 7, lines 333-335: could the citation be changed to show how FGD applies to social issues instead of products, services, and system features?
Page 7, line 338: how do the authors know that the FGDs provided richer data than interviews when they did not conduct interviews?
Page 7, line 343: this should go into the limitation section.
The section on recruitment should also include information on how participants were recruited, not just from where. How did the participants learn about the study? Were flyers distributed? Notices put up at the library? Were interested participants asked to email the researchers to express interest? The process of recruitment should be included here.
Page 7, line 349: The nature of purposive sampling is to select information rich cases to get in-depth information about a certain phenomenon. To say that purposive sampling introduces selection bias is incorrect. It may lead to self-selection of participants who wish to participate, but purposive sampling and qualitative research in general cannot be viewed with the same lens as quantitative methods. As with the last version of this manuscript that mentioned the need for quantitative verification of the study’s findings, in this version, mentioning the need to increase generalizability/representativeness with a larger sample at the very beginning of the methods section, as well as selection bias, makes it seem as though the researchers are not convinced about the method that they have chosen for this study or they think that qualitative research is in some way lacking when compared to quantitative methods. I also get the impression that the authors are trying to frame this study as a necessary step towards an “actual” study. There are references to this study looking at only this specific sample but laying the groundwork for future studies, the need to be more representative, the “limited” sample of 30 etc. all of which seem to be downplaying the significance of this study. There is nothing wrong in doing a qualitative study before designing a larger survey based on the findings, for example, but as a reader, I get the impression that the authors are unconvinced about qualitative methods which is not a nice impression to leave a reader with. If needed, maybe the authors could ask a qualitative researcher to take a look at the manuscript.
This section should also explain how/why the authors decided on 30 participants. It is important to state the rationale for sample size.
Page 8, lines 360-362: the last past of the sentence (“…rather than representing the views of all older adults”) is not possible anyway so does not need to be mentioned.
Page 8, lines 369-373: this should go into the limitations/recommendations section.
Could Table 1 also include something about the level of climate related knowledge since the sample was stratified by climate awareness?
Page 9, line 438: please cite Bryman.
Page 9, line 441: in qualitative research, researchers look for dependability and trustworthiness of data instead of reliability.
What kind of triangulation methods did the researchers engage in?
Page 10, lines 445-446: this is an abrupt transition from familiarization to initial codes.
Page 10, line 469: Please label it “Theme 1: Suggestions for…” for clarity and also because it is easier to refer to specific themes in the discussion section. Same for the other 3 themes.
Page 11, lines 521 to 529: this should be part of discussion. The results should only include findings.
Page 12, lines 567-570: this should be part of discussion.
Page 13, lines 624-630: this should be part of discussion.
Page 14, lines 651-653: this should be part of discussion.
Page 14, lines 671-675: this is a repetition of lines 666-669.
General: at the end of all the sub-themes there are recommendations for best practices that should go into the discussion section.
Page 16, line 770: 4.3.6 – this section could use some more quotes, especially the last 2 paragraphs on encouragement and attraction. It is not clear in these two whether the authors are making general statements or if the participants mentioned these points.
Page 19, line 927: knowledge transfer usually takes place from older to younger generations, so this statement needs a little clarification if it is trying to say that current climate-related knowledge is shared by younger generations with older generations.
While the discussion section is better, at least 1-2 limitations of the study must still be acknowledged.
Minor editorial issues:
Page 2, line 45: Sentence beginning with “In 2024…”
Page 3, line 139: needs revision.
Page 7, line 334: there’s a “175” in the middle of the sentence.
Page 10, line 461: “reveals how”
Page 17, line 790 “may be”
Author Response
Dear reviewer,
We sincerely thanks for your detailed and insightful feedback, which has greatly improved the clarity and quality of our manuscript. Below, we provide a point-by-point response to the comments and outline the corresponding revisions. All page and line numbers refer to the revised manuscript.
- Introduction: Indicate the study was undertaken in the UK with London-based older adults (line 68).
Reviewer Comment: It would be good to mention somewhere that this study was undertaken in the UK with London-based older adults—maybe in the paragraph beginning line 68.
Response: We have incorporated this suggestion into the introduction.
Revision: Added: "This study, conducted in the UK, focuses on older adults residing in London. Through focus group interviews, it delves deeply into their perceptions, behaviours, and recommendations regarding climate action." (Page 2, line 68)
- Page 3, line 120: Citation needed.
Reviewer Comment: Citation needed.
Response: A citation has been added to support this statement.
Revision: Added: "Research highlights that older adults often show a strong environmental concern, commonly described as prosocial behaviour, when addressing climate change [21]." (Page 3, line 120)
Reference Added:
Cutler, J.; Nitschke, J.P.; Lamm, C.; Lockwood, P.L. Older adults across the globe exhibit increased prosocial behavior but also greater in-group preferences. Nature Aging 2021, 1, 880–888.
- Clarify connection between pro-social behaviour and climate action (lines 120-128).
Reviewer Comment: I was unable to make the connection between pro-social behaviour and climate action, especially in light of the cited article on socio-emotional selectivity theory. Could the authors add a sentence to link these?
Response: We clarified the connection by linking generativity and legacy concerns with prosocial behaviours related to climate action.
Revision:
"Research highlights that older adults often exhibit a strong environmental concern, commonly characterised as prosocial behaviour, in the context of addressing climate change. This prosocial outlook is rooted in their altruistic tendencies and is often guided by a focus on emotionally meaningful relationships and deeper connections with family and friends. Their concern for future generations, driven by a sense of legacy and generativity, further motivates their engagement in actions that benefit others and the environment. These prosocial tendencies are reflected in behaviours such as fostering social bonds, caring for others, and offering support within close-knit social networks. This generational concern can extend to climate action, where older adults may adopt low-carbon behaviours like using energy-saving technologies or participating in sustainable travel options to ensure a better future for their descendants." (Page 3, lines 120-132)
- Page 4, line 149: Citation needed for UNDESA.
Reviewer Comment: Citation needed for UNDESA.
Response: The relevant citation has been added.
Revision: Added: "United Nations Department of Economic and Social Affairs (UNDESA), Population Division. World population ageing 2019: Highlights (ST/ESA/SER.A/430), 2019. Accessed: 2023-12-09."
- Page 4, line 152: Citation needed for Age UK.
Reviewer Comment: Citation needed for Age UK.
Response: Two supporting citations from Age UK and related reports have been added.
Revision:
"Similarly, Age UK’s Parliamentary Briefing: The Energy Crisis and Supporting Older People this Winter (2021) stresses the need for targeted support to help older adults navigate energy crises and transition to sustainable energy practices [32]. The Healthy Ageing in a Changing Climate report underscores the importance of involving older adults in climate action through community education and improved resource access, advocating for inclusive, age-friendly strategies to empower older populations in addressing climate challenges [33]." (Page 4, line 156)
References Added:
- UK, A. Parliamentary Briefing: The Energy Crisis and Supporting Older People this Winter, 2021. Available at: https://www.ageuk.org.uk/globalassets/age-uk/documents/reports-and-publications/parliamentary-briefings/energy-crisis-supporting-older-people-briefing.pdf
- Woolrych, R.; Haq, G.; Latter, B. Healthy Ageing in a Changing Climate: Creating Inclusive, Age-Friendly, and Climate Resilient Cities and Communities in the UK; PlaceAge Project, 2023. Available at:https://placeage.org/publications
- Page 4, line 154: Citation needed for Green Homes Grant.
Reviewer Comment: Citation needed for Green Homes Grant.
Response: The appropriate citation has been added.
Revision: Added:
- Government, U. Green Homes Grant Scheme: Improving Energy Efficiency for Homes, 2020. Available at: https://www.gov.uk/greenhomesgrant
- Page 4, line 165: Citations needed for “several studies.”
Reviewer Comment: Citations needed for “several studies.”
Response: Supporting references have been added to substantiate this statement.
Revision:
"Older adults have largely been overlooked in climate change discussions, creating a critical knowledge gap regarding their role in addressing the climate crisis [13]. The carbon footprint of older people is closely linked to their lifestyle [35], consumption patterns [36], and health needs [37], with several studies offering different perspectives on this issue." (Page 4, line 170)
References Added:
- Ivanova, D.; Stadler, K.; Steen-Olsen, K.; Wood, R.; Vita, G.; Tukker, A.; Hertwich, E.G. Environmental impact assessment of household consumption. Journal of Industrial Ecology 2016, 20, 526–536.
- Long, Y.; Feng, J.; Sun, A.; Wang, R.; Wang, Y. Structural Characteristics of the Household Carbon Footprint in an Aging Society. Sustainability 2023, 15, 12825.
- Fuchs, D.A.; Lorek, S. Sustainable consumption governance: A history of promises and failures. Journal of Consumer Policy 2005, 28, 261–288.
- Page 4, line 184: Clarify how “lifestyle” is distinct from other points mentioned; provide an example.
Reviewer Comment: It is not clear how “lifestyle” is separate from the other points mentioned in the paragraph. Maybe give an example.
Response: A specific example has been added to clarify the distinction.
Revision:
"4. Lifestyles: Older adults’ established habits and routines can pose challenges to adopting new, long-term strategies for reducing carbon emissions. For example, older adults in Europe tend to rely more on personal vehicles for daily errands and social visits, particularly in rural or suburban areas where public transport options are limited [40]. In contrast, older adults in Asia are more likely to use public transport or walk for similar activities, especially in urban environments with well-developed infrastructure [41]. These differences highlight how individual behaviors are shaped by cultural and regional contexts, underscoring the need for further exploration of these patterns to design tailored, low-carbon interventions [17]." (Page 4, line 190)
- Page 5, line 204: Correct “life cycle perspective” to “life course perspective.”
Reviewer Comment: The authors have stated “life cycle perspective” but cited “life course perspective.” These are two different things. Life course perspective is more appropriate in the context of this discussion.
Response: The terminology has been corrected to “life course perspective.”
Revision: Updated throughout the manuscript to ensure accuracy and consistency.
- Page 5, line 213: Replace “similar” with “varied” to better reflect changes older adults have experienced.
Reviewer Comment: I am not sure it is accurate to say that older adults have lived through similar social and environmental changes as both climate change and technological change at current levels are unprecedented. Maybe replace “similar” with “varied.”
Response: The wording has been updated to “varied” to reflect the reviewer’s suggestion.
Revision:
"Secondly, having lived through varied social and environmental changes, older people share a collective understanding of how to adapt to shifting circumstances." (Page 5, line 223)
- Page 5, line 232: Replace “senior” with “older.”
Reviewer Comment: “Senior” is also considered ageist when used to refer to an older person. Replace it with “older.”
Response: The term “senior” has been replaced with “older” throughout the manuscript.
- Page 5, line 239: Reframe the sentence to exclude “should.”
Reviewer Comment: Could the authors reframe this sentence to exclude “should”? Things that should be done usually go into recommendations at the end of the paper.
Response: The sentence has been rephrased to align with the reviewer’s recommendation.
- Page 6, line 248: Recommendations should go in the discussion section.
Reviewer Comment: Here as well, this is a recommendation that does not fit in here.
Response: Recommendations have been relocated to the discussion section to ensure proper placement.
- Page 6, line 263: Same issue; move recommendations to discussion.
Reviewer Comment: These should go at the end of the paper or in the discussion.
Response: Similar to the previous point, these recommendations have been consolidated and relocated to the discussion section.
- Page 7, line 325: Replace “study 1” with more appropriate phrasing.
Reviewer Comment: Why study 1?
Response: The phrasing has been revised to avoid confusion.
Revision: Updated to “this study.”
- Page 7, lines 333–335: Change citation to show how FGD applies to social issues.
Reviewer Comment: Could the citation be changed to show how FGD applies to social issues instead of products, services, and system features?
Response: A new reference demonstrating the use of FGDs in addressing social issues has been added.
Revision:
Added citation:
- Kitzinger, J. Introducing Focus Groups. BMJ 1995, 311(7000), 299-302. DOI: 10.1136/bmj.311.7000.299
- Page 7, line 338: Clarify why FGDs provided richer data without interviews.
Reviewer Comment: How do the authors know that the FGDs provided richer data than interviews when they did not conduct interviews?
Response: The section has been clarified to avoid unjustified comparisons between FGDs and interviews.
Revision:
"The discussions enabled participants to inspire one another, leading to new ideas or perspectives. The group setting facilitated dynamic interactions, which are less likely to occur in one-on-one settings. While individual interviews were not conducted in this study, the focus group discussions allowed researchers to observe commonalities and differences among participants, providing a comprehensive understanding of their thoughts and motivations." (Page 7, line 345)
- Page 7, line 343: Move content to the limitations section.
Reviewer Comment: This should go into the limitation section.
Response: The relevant content has been summarized and added to the limitations section in the discussion.
- Recruitment process clarification (Page 8, line 362).
Reviewer Comment: Include more details on how participants were recruited, not just from where.
Response: Details about the recruitment process have been added.
Revision:
"The main participants were recruited from the Brunel Older People's Reference Group (BORG). Invitations were sent to BORG members via email, explaining the purpose of the study and inviting them to participate in focus group discussions. BORG members were also encouraged to share these invitations with their networks. Additionally, recruitment efforts extended to local churches and the library in West London, where posters were displayed on community boards and informational cards were distributed to older adults visiting these locations. Interested participants were asked to express their willingness to participate by emailing or texting the researchers, allowing the team to schedule sessions at mutually convenient times." (Page 8, line 362)
- Page 7, line 349: Avoid calling purposive sampling “biased.”
Reviewer Comment: Purposive sampling should not be described as introducing bias; it is inappropriate to evaluate qualitative sampling with the same lens as quantitative methods.
Response: The description of purposive sampling has been revised to avoid mischaracterizing it as introducing bias.
Revision:
"A purposive sampling strategy was used to recruit older participants, which is suitable for situations requiring an in-depth understanding of a specific group’s views, experiences, or behaviours [73]. This approach focuses on selecting information-rich cases to provide deeper insights into the phenomenon under study, rather than aiming for statistical representativeness." (Page 8, line 352)
- Page 7, line 349: Clarify the rationale for choosing 30 participants.
Reviewer Comment: The authors should explain how/why they decided on 30 participants.
Response: The rationale for the sample size has been clarified.
Revision:
"The sample size of 30 participants was chosen to ensure a balance between obtaining diverse perspectives and achieving data saturation, where no new themes emerge from additional data collection." (Page 8, line 384)
- Page 8, lines 360–362: Avoid stating that views represent “all older adults.”
Reviewer Comment: The statement “…rather than representing the views of all older adults” is not possible anyway, so does not need to be mentioned.
Response: The sentence has been revised to align with the reviewer’s suggestion.
Revision:
"The aim of this study was to explore older adults’ perceptions and behaviours regarding climate change, focusing on a group capable of engaging in in-depth discussions. This approach sought to provide a detailed understanding of the low-carbon behaviours and climate change perceptions of this specific group, laying the groundwork for future research." (Page 8, line 375)
- Page 8, lines 369–373: Move to limitations/recommendations section.
Reviewer Comment: This should go into the limitations/recommendations section.
Response: The content has been moved and incorporated into the limitations and recommendations section of the discussion.
- Table 1: Add information on climate knowledge levels.
Reviewer Comment: Could Table 1 also include something about the level of climate-related knowledge since the sample was stratified by climate awareness?
Response: A section on self-assessed climate knowledge levels has been added to Table 1.
Revision:
Participants were asked to self-assess their level of climate knowledge, which was categorized into four tiers:
- No Knowledge: Participants had no understanding of or exposure to concepts related to climate change or associated terminology.
- Basic Knowledge: Participants had some awareness of climate change but were unable to explain its causes or related policies in depth.
- Moderate Knowledge: Participants demonstrated a basic understanding of climate change concepts and its primary causes.
- Advanced Knowledge: Participants were able to clearly articulate the concept of climate change, its societal impacts, and were familiar with related policies and low-carbon lifestyle practices.
- Page 9, line 438: Cite Bryman.
Reviewer Comment: Please cite Bryman.
Response: A citation to Bryman has been added in the relevant section.
Revision:
- Bryman, A. Social Research Methods. 5th ed.; Oxford University Press, 2016.
- Page 9, line 441: Replace “reliability” with “dependability and trustworthiness.”
Reviewer Comment: In qualitative research, researchers look for dependability and trustworthiness of data instead of reliability.
Response: The terminology has been revised to align with qualitative research principles.
Revision:
"In qualitative research, dependability and trustworthiness are prioritized to ensure rigorous data analysis."
- Clarify triangulation methods (Page 10, lines 445–446).
Reviewer Comment: What kind of triangulation methods did the researchers engage in?
Response: Details on the triangulation methods have been provided.
Revision:
"This study employed Investigator Triangulation and Data Source Triangulation. For investigator triangulation, two researchers independently reviewed and analyzed the data, followed by team discussions to consolidate themes and minimize potential biases. Data source triangulation was achieved by recruiting participants from diverse settings, including libraries, churches, and older people's reference groups, ensuring a range of climate knowledge levels and backgrounds." (Page 10, line 448)
- Page 10, lines 445–446: Address abrupt transition from familiarization to coding.
Reviewer Comment: This is an abrupt transition from familiarization to initial codes.
Response: The transition has been smoothed to improve readability.
Revision:
"The data analysis followed Braun and Clarke’s six-phase framework: familiarising with the data, generating initial codes, searching for themes, discussing and validating themes with co-researchers, defining and naming the themes, and producing the final report [75]. The analysis began with an in-depth review of the transcribed focus group discussions to gain familiarity with participants’ insights and interactions. From this, initial codes were generated to capture recurring ideas and patterns within the data. For instance, 'frugality as a cultural norm' was coded based on participants frequently referencing resource-saving habits shaped by past economic hardships. Similarly, 'intergenerational dialogue' emerged from discussions about sharing climate-related knowledge with younger family members." (Page 10, line 459)
- Page 10, line 469: Label themes for clarity.
Reviewer Comment: Please label it “Theme 1: Suggestions for…” for clarity and also because it is easier to refer to specific themes in the discussion section.
Response: All themes have been labeled for clarity and consistency.
30–32. Move findings to the discussion section (Pages 11–14).
Reviewer Comment: Pages 11, 12, and 13 contain content that belongs in the discussion section.
Response: The findings have been consolidated into the discussion section, ensuring a clear distinction between results and interpretation.
- Page 14, lines 671–675: Remove repetition.
Reviewer Comment: This is a repetition of lines 666–669.
Response: The repetition has been removed.
- Page 16, line 770: Add quotes to sections on encouragement and attraction.
Reviewer Comment: This section could use more quotes, especially the last two paragraphs on encouragement and attraction.
Response: Additional participant quotes have been added to clarify that these are participant-driven insights.
Revision:
See updated section with participant quotes under Page 16, line 764.
In the focus groups, participants discussed various methods to remind people about the impact of their actions and motivate them to engage in more environmentally responsible behaviour. Older adults frequently mentioned reward and penalty systems as formal mechanisms that can regulate behaviour. These systems not only influence individuals through incentives and sanctions but also provide information that evaluates specific actions. Such information serves as a subtle reminder to reflect on whether their behaviour aligns with environmental standards. For example, older adults referred to news reports showing images of the destruction of the Amazon rainforest or homeless animals, which served as "soft nudges," helping them realise the environmental consequences of their daily choices. One participant remarked, “When I see those pictures of animals losing their habitats, it makes me think—what have I done today that could be hurting the planet? ” (Female, 68). Rather than delivering an emotionally charged message, these reminders gently prompt individuals to reconsider their behaviour, fostering self-awareness in a non-intrusive way.
Beyond reminders, participants highlighted the importance of warnings that emphasize the potential consequences of inaction. Warnings, compared to gentle nudges, deliver a more serious message aimed at capturing attention and informing individuals of the risks associated with their current behaviour. These warnings can evoke stronger emotional responses and create a sense of urgency or responsibility. For instance, one participant stated, “If you tell people about fines for leaving the heating on too high, they’ll listen. But just saying it’s about saving the planet? Not everyone cares as much” (Male, 72). This reflects the perception that monetary penalties may be more effective than moral appeals in motivating behavioural changes.
In contrast to warnings, rewards offer positive reinforcement. Older adults expressed interest in financial incentives, such as subsidies for switching to renewable energy sources. These tangible rewards could help encourage environmentally conscious behaviour. However, rewards need not always be financial. Verbal encouragement and positive reinforcement, such as acknowledging good practices or providing feedback through informational cues, can serve as powerful motivators. One participant shared, “Even if it’s just a thank-you letter for participating in a recycling program, I’d feel good about it. It makes you want to do more” (Female, 74). Another noted, “When someone says, ‘You’ve saved energy this month,’ it feels like you’ve achieved something. That’s encouragement enough” (Male, 65). Encouragement builds a supportive environment by praising positive actions, which helps sustain long-term engagement with low-carbon practices.
Finally, participants discussed the role of attraction as a motivational strategy, with the goal of drawing attention through visually engaging or creatively designed content. Rather than directly prompting action, attraction works by sparking curiosity and stimulating exploration, encouraging individuals to learn more about climate change. One participant noted, “I love those colourful infographics that show you how small actions add up to big changes—they’re so much easier to understand than just numbers” (Female, 71). Another added, “If it’s a documentary with a good story, I’ll watch it. But if it’s too technical, I lose interest” (Male, 67). By engaging people through appealing visuals or compelling stories, attraction can lead to greater awareness and, ultimately, more sustained involvement in climate action.
- Page 19, line 927: Clarify knowledge transfer in intergenerational dialogue.
Reviewer Comment: Knowledge transfer usually takes place from older to younger generations, so this statement needs clarification.
Response: I changed the intergenerational interaction in the discussion section.
Revision:
"Family and friends play an important role in shaping their understanding of climate issues, showing the value of intergenerational conversations in driving climate action." (Page 19, line 919)
"Moreover, participants mentioned the motivation of protecting the earth for the next generation. Intergenerational dialogue also emerged as a powerful tool, with older adults sharing stories and practical advice with family members." (Page 20, line 930)
- Add study limitations (Discussion).
Reviewer Comment: While the discussion section is better, at least 1–2 limitations of the study must still be acknowledged.
Response: Limitations have been added to the discussion.
Revision:
"This study has certain limitations that should be acknowledged. First, the sample size was limited to 30 participants, all residing in West London. While this provided valuable perspectives and rich data, the findings may not fully capture the diversity of older adults’ views and behaviours across different regions or socioeconomic contexts. Expanding the sample size and including participants from a broader geographic range would enhance the representativeness of future studies.
Additionally, this study focused on older adults who were able to attend group discussions, which may have excluded those aged 85 and above or individuals with mobility or language barriers. Future research could adopt tailored methods, such as one-on-one interviews or flexible, home-based approaches, to better address the needs of these underrepresented groups." (Page 22, line 1042)
- Page 2, line 45: Revise “In 2024…” sentence.
Reviewer Comment: Sentence beginning with “In 2024…” needs revision.
Response: The sentence has been revised for clarity.
Revision:
"A 2024 study found that younger people are more likely than older adults to believe in the attainability of the net-zero target."
- Page 3, line 139: Needs revision.
Reviewer Comment: Revise sentence on heatwaves in the UK.
Response: The sentence has been revised for clarity and conciseness.
Revision:
"In the UK, heatwaves pose significant health risks to older adults, with extreme temperature events contributing to excess mortality among this vulnerable population."
Reviewer 2 Report
Comments and Suggestions for Authors
The authors have incorporated the comments, and the manuscript is ready to be published by the journal.
Author Response
Dear Reviewer,
Thank you for your positive feedback and for acknowledging the incorporation of your comments. We greatly appreciate your thoughtful and constructive suggestions during the review process, which have significantly improved the clarity and quality of the manuscript.
We are grateful for your recommendation and look forward to the next steps in the publication process.
Sincerely,
Qing